# Leveraging Beneficial Off-Target Effects of Live-Attenuated Rotavirus Vaccines

**DOI:** 10.3390/vaccines10030418

**Published:** 2022-03-10

**Authors:** Prisca Benedicto-Matambo, Julie E. Bines, Chikondi Malamba-Banda, Isaac T. Shawa, Kayla Barnes, Arox W. Kamng’ona, Daniel Hungerford, Kondwani C. Jambo, Miren Iturriza-Gomara, Nigel A. Cunliffe, Katie L. Flanagan, Khuzwayo C. Jere

**Affiliations:** 1Virology Research Group, Malawi-Liverpool-Wellcome Trust Clinical Research Programme, Blantyre 312225, Malawi; pbenedicto@mlw.mw (P.B.-M.); cmalamba@mlw.mw (C.M.-B.); ishawa@medcol.mw (I.T.S.); kbarnes@broadinstitute.org (K.B.); awkamngona@kuhes.ac.mw (A.W.K.); kondwani.jambo@lstmed.ac.uk (K.C.J.); 2Centre for Global Vaccine Research, Institute of Infection, Veterinary and Ecological Sciences, University of Liverpool, Liverpool L69 7BE, UK; danhungi@liverpool.ac.uk (D.H.); miturrizagomara@path.org (M.I.-G.); nigelc@liverpool.ac.uk (N.A.C.); 3Department of Medical Laboratory Sciences, Faculty of Biomedical Sciences and Health Professions, College of Medicine, Kamuzu University of Health Sciences, Blantyre 312225, Malawi; 4Enteric Diseases Group, Murdoch Children’s Research Institute, Department of Gastroenterology and Clinical Nutrition, Royal Children’s Hospital and Department of Paediatrics, The University of Melbourne, Parkville, VIC 3052, Australia; julie.bines@mcri.edu.au; 5Department of Biological Sciences, Academy of Medical Sciences, Malawi University of Science and Technology, Blantyre 312225, Malawi; 6Harvard School of Public Health, Boston, MA 02115, USA; 7Department of Biomedical Sciences, Faculty of Biomedical Sciences and Health Profession, College of Medicine, Kamuzu University of Health Sciences, Blantyre 312225, Malawi; 8NIHR Health Protection Research Unit in Gastrointestinal Infections, University of Liverpool, Liverpool L69 7BE, UK; 9Department of Clinical Sciences, Liverpool School of Tropical Medicine, Liverpool L3 5QA, UK; 10Centre for Vaccine Innovation and Access, Program for Appropriate Technology in Health (PATH), 1218 Geneva, Switzerland; 11School of Medicine, University of Tasmania, Hobart, TAS 7005, Australia; katie.flanagan@ths.tas.gov.au; 12School of Health and Biomedical Science, Royal Melbourne Institute of Technology (RMIT), Bundoora, VIC 3083, Australia; 13Department of Immunology and Pathology, Monash University, Melbourne, VIC 3004, Australia

**Keywords:** rotavirus, off-target effects, neonatal, live attenuated, RV3-BB, epigenetic modulation

## Abstract

Following the introduction of live-attenuated rotavirus vaccines in many countries, a notable reduction in deaths and hospitalisations associated with diarrhoea in children <5 years of age has been reported. There is growing evidence to suggest that live-attenuated vaccines also provide protection against other infections beyond the vaccine-targeted pathogens. These so called off-target effects of vaccination have been associated with the tuberculosis vaccine Bacille Calmette Guérin (BCG), measles, oral polio and recently salmonella vaccines, and are thought to be mediated by modified innate and possibly adaptive immunity. Indeed, rotavirus vaccines have been reported to provide greater than expected reductions in acute gastroenteritis caused by other enteropathogens, that have mostly been attributed to herd protection and prior underestimation of rotavirus disease. Whether rotavirus vaccines also alter the immune system to reduce non targeted gastrointestinal infections has not been studied directly. Here we review the current understanding of the mechanisms underlying off-target effects of vaccines and propose a mechanism by which the live-attenuated neonatal rotavirus vaccine, RV3-BB, could promote protection beyond the targeted pathogen. Finally, we consider how vaccine developers may leverage these properties to improve health outcomes in children, particularly those in low-income countries where disease burden and mortality is disproportionately high relative to developed countries.

## 1. Introduction

Diarrhoea, pneumonia and malaria remain the leading causes of death among children under the age of five years [1]. Approximately five million children fail to reach the age of five every year due to treatable and vaccine preventable diseases [1]. Although efforts have been accelerated to reduce the rate of under-five mortality worldwide, an imbalance still exists between low- and high-income countries (HIC) [1]. In Sub-Saharan Africa, infant mortality remains high, and one in every 13 children fail to reach their fifth birthday compared to their peers in HIC [1].

Notably, diarrhoeal disease associated with rotavirus infection accounts for an estimated 128,500 under-five deaths annually, despite increasing rollout of rotavirus vaccines [2,3]. Rotavirus vaccines are key interventions to counteract the devastating effects of rotavirus-associated diarrhoea [4]. To date, over 100 countries have incorporated rotavirus vaccines into their national immunisation programmes [5,6]. Following implementation of rotavirus vaccination, reduction in diarrhoea-associated deaths has been observed in many countries [7]. Observational studies conducted in both high income and middle-income settings suggest that rotavirus vaccines may also provide protection against severe acute gastroenteritis not associated with rotavirus [8,9,10]. Further understanding of the mechanisms through which new and existing vaccines induce broad-based immunity against a range of pathogens would be of considerable value to high disease-burdened regions.

Both epidemiological and immunological studies suggest that certain vaccines exhibit immuno-modulatory functions that induce beneficial off-target effects against unrelated pathogens. Whilst limited data are available to explain this phenomenon, non-specific effects are in part thought to originate from epigenetic reprogramming of innate cells that subsequently lead to altered innate memory. Strong evidence on non-specific effects has been reported from live-attenuated vaccines, notably Bacille Calmette Guérin (BCG) and measles vaccines [11,12,13,14]. It is uncertain whether similar effects are applicable to rotavirus vaccines. We postulate that live-attenuated rotavirus vaccines administered orally at birth might induce beneficial non-specific effects, and confer protection to unrelated pathogens, and have therefore reviewed the available literature in this area.

## 2. Off-Target Effects of Vaccines

Contrary to the traditional view that vaccines induce specific immunity to a specific pathogen, evidence has emerged over the past decades that suggests vaccines can also induce protection against non-targeted pathogens. This phenomenon has been termed ‘non-specific effects’, ‘heterologous effects’ or ‘off-target effects’ of vaccination, and from herein this article will use the latter term. Both epidemiological and immunological studies have demonstrated protective off-target effects from live vaccines whereby they reduce susceptibility to non-targeted pathogens. By contrast, non-live vaccines such as whole cell Diphtheria-Tetanus-Pertusis (DTP) may have been associated with negative effects such as slightly increased morbidity and mortality from non-targeted pathogens, particularly observed in low income-high mortality settings [12,15,16]. The World Health Organisation (WHO) initially expressed reservations regarding the existence of vaccine off-target effects, largely due to data originating from observational non-randomised studies that were open to potential bias [17]. However, a systematic review conducted by WHO confirmed that measles and BCG vaccination in infancy are associated with a significantly reduced all-cause mortality from non-targeted pathogens [13], and several randomised studies have consolidated evidence that support vaccine-induced off-target effects [13,15,17,18]. To date, some evidence for off-target effects have emerged from randomised studies of BCG, measles and oral polio vaccine (OPV), with more recent evidence emerging for the live-attenuated salmonella Ty21a vaccine [18].High-income countries such as UK and USA and low-income countries including Guinea-Bissau, The Gambia, Senegal and Sudan have all reported off-target effects of vaccination from both randomised and non-randomised studies [12,19,20,21,22].

## 3. Mechanisms of Off-Target Effects of Vaccination: Major Lessons from Bacille Calmette Guérin

Despite the concept of off-target effects of vaccines attracting attention from scientists worldwide, the precise immunological mechanisms associated with these effects remain unclear. Nevertheless, several vaccine effect studies have investigated both innate and adaptive immune components as potential mediators of off-target effects. BCG is a widely used anti-tuberculosis vaccine and a well-studied model of vaccine-induced off-target effects. BCG-induced off-target effects have been widely investigated using murine models by comparing immune responses following heterologous antigen challenge between BCG naïve and BCG-vaccinated mice. Earlier studies by Spencer et al. (1976) reported increased capacity of BCG-stimulated macrophages in mice in reducing viral titres of influenza A virus compared with controls. A study by Starr et al. further demonstrated that a single dose of BCG vaccine administered intradermally or intraperitoneally protected newborn mice from herpes simplex virus type 2 in a controlled murine model [23]. It has been suggested that shared cross-reactive epitopes between BCG and unrelated pathogens might explain the observed non-specific protective responses upon secondary exposure to the unrelated pathogens [11,24,25]. However, solid evidence is required to directly link BCG signatures to the multitude of unrelated pathogens for which off-target effects have been observed [11].

A further hypothesis is that BCG induces off-target protection via the production of cytokines that alter the inflammatory environment, thereby enabling activation of pre-existing effector or memory T cells, B cells or natural killer cells in what is called bystander effects [11]. It has been further suggested that BCG induces off-target protection via innate cell-mediated production of cytokines that accelerate T follicular helper (T_FH_) cell development and B-cell maturation, thus enhancing a broad range of antibody responses against an array of pathogens [26,27,28,29]. Since the infant immune system is not fully developed and sufficiently experienced to generate broad pre-existing T and B cell memory, these bystander effects seem unlikely to play a major role in BCG-induced off-target effects in this age group [11,30]. 

Additionally, Toll-like receptor 8 (TLR8) engagement with BCG is also implicated in potentiating off-target effects. BCG engagement with TLR8, which is preferentially expressed on dendritic cells, monocytes and macrophages is known to induce high levels of cytokines including interleukin 12 (IL-12) and tumour necrosis factor-alpha (TNF-α). Induction of off-target effects using this pathway appears plausible as IL-12 signalling is thought to drive differentiation and function of T_FH_ cells that are crucial for providing help to B cells, subsequently augmenting antibody-mediated immune responses. Brook et al. (2020) proposed emergency granulopoiesis as another possible mechanism linked to BCG protection against unrelated pathogens. Using a neonatal mouse model, the study demonstrated that BCG vaccination led to emergency granulopoiesis via upregulation of granulocyte colony stimulating factor (G-CSF), and that emergency granulopoiesis was associated with rapid neutrophil expansion. These findings imply that BCG-induced granulopoiesis could potentially confer protection against microbial-induced sepsis. Similar observations were made in human neonatal studies, although the effect observed was rapid and short-lived (24 h) [31]. If extensively studied, granulopoiesis could lessen sepsis-related deaths in resource constrained settings where more than half of child mortality occurs within the neonatal period [31,32].

The concept of trained immunity is one of the most plausible mechanisms of BCG vaccine-induced off-target effects. It is based on the understanding that innate immune cells can undergo epigenetic changes following exposure to certain vaccines or microbial components [11,15]. This results in long-term changes in innate immunity i.e., innate immune memory, which is at odds with prevailing thinking that has historically restricted immunological memory to the adaptive components of the immune system such as B and T lymphocytes [33,34,35,36,37,38] Indeed, there is evidence that innate immune memory is an inherent trait in plants, animals and invertebrates and is used as a mechanism of protection [39,40]. An in vitro experimental model where primary monocytes are stimulated with potential heterologous inducers such as beta-glucan or BCG and then re-stimulated with agents such as lipopolysaccharide that act as a proxy of heterologous pathogens has been used to demonstrate innate immune training. Such studies have shown that innate immune cells can undergo metabolic, functional and epigenetic modulation leading to the acquisition of long-lasting memory responses to subsequent unrelated stimuli [38].

Advancements in genome wide studies, proteomics, transcriptomics, metabolomics and epigenetics have allowed discovery of signalling pathways and signatures that may be involved in innate cell training [11,41]. Molecular pathways such as protein kinase B/mechanistic target of rapamycin (Akt/mTOR), phosphoinositide/protein kinase B (PI3K/Akt), nucleotide-binding oligomerisation domain 2 (NOD2)-pattern receptor signalling, histone modification and NF-ĸB transcription factors have been highlighted as key inducers of trained immunity in monocytes. Expression of pattern recognition receptors (PRR) including TLR2, TLR3 and TLR4 as well as costimulatory molecules such as CD40, CD80 are upregulated during innate cell training. It is therefore not surprising that many studies have reported upregulation of inflammatory cytokine genes, such as IL-1β TNF-α, IL-6, which are involved in a wide range of heterologous responses [14,42,43,44,45].

## 4. Off-Target Effects from Live and Non-Live Vaccines

Off-target effects induced by vaccines are generally characterised as beneficial or deleterious depending on the nature of a particular vaccine. Beneficial off-target effects are commonly associated with live vaccines and are thought to confer protective immunity. In contrast, deleterious effects have been reported from epidemiological studies on some inactivated vaccines that are linked with increased all-cause mortality [16,19].

Off-target effects were first reported in a large cohort study conducted in Guinea-Bissau two decades ago which demonstrated increased mortality in children that received whole cell diphtheria-tetanus-pertussis (DTP) vaccine [21,22,46,47,48,49,50]. These effects are thought to last until the next vaccine is administered and would become beneficial if the next vaccine was a live vaccine [16]. A large USA study by Bardenheier et al. (2017) spanning 10 years involving 311,663 children showed a lower risk of non-targeted infections when the last vaccine was live compared to inactivated (HR 0.5, 95% confidence interval (CI) 0.43–0.57) [51]. However, the concept of deploying sequential vaccine administration to counteract negative off-target effects remains speculative considering that infants in high endemic settings are likely to be exposed to environmental or maternal pathogens, including organisms such as norovirus, enteric adenoviruses and campylobacter before the next vaccine is administered [52,53,54]. Whether or not these enteric pathogens would similarly enhance the developing infant immune system remains unestablished, but a similar immune-modulatory effect might not be surprising considering that these are also live organisms. Studies that could mechanistically investigate why certain vaccines appear to induce off-target effects irrespective of prevailing pathogen exposure or indeed the possibility of vaccine-pathogen co-immunomodulatory impact in high endemic countries would, to some extent, provide a better understanding on the plausibility of off-target effects in these settings. The WHO recommends concomitant administration of rotavirus vaccine with DTP [55,56]. The impact of this co-administration on off-target effects of each vaccine requires attention in settings that are implementing these guidelines.

BCG-induced off-target effects were implicated in the reduction in all-cause mortality from six controlled studies conducted in the USA in the 1940–1950s [12,15,50]. Similar observations were made in a randomised controlled trial in low-birth-weight infants in Guinea-Bissau that reported a lower mortality rate ratio following administration of the BCG vaccine as compared to unvaccinated controls [12]. Combined findings from two randomised trials in Guinea-Bissau reported a reduced neonatal mortality ratio of 0.52 (0.33–0.82) following a birth dose of BCG as compared to no vaccine. These findings have profound implications as more than half of infant mortality in resource limited settings occurs during the neonatal period [15,16,50].

Apart from inducing specific immunity against targeted pathogens, non-live vaccines are thought to induce deleterious effects that lead to increased all-cause mortality. Some studies in high mortality settings in Africa and Asia on infants vaccinated with DTP have been linked with an increase in all-cause childhood mortality, particularly in girls [57]. A 4.33-fold increase in mortality was observed compared to controls among low birthweight neonates in Guinea randomly assigned to receive BCG vaccine at birth followed by DTP vaccine two months later. Since mortality rate is thought to be determined by the last vaccine administered, the increased mortality observed in this study was attributable to DTP vaccination [15,16]. Other vaccines associated with negative off-target effects include single hepatitis B vaccine, Pentavalent vaccine (DTP, hepatitis B virus, *Haemophilus influenza* type b), inactivated polio vaccine, H1N1 influenza vaccine and the malaria vaccine RTS, S/AS01 [13,57,58,59,60,61,62,63]. In the context of the malaria vaccine, it is clear that there is no randomised controlled trial data, hence interpretation is based on observations. Clearly, non-live vaccines are important in inducing specific immunity to their targeted pathogens, but the reported negative effects from some studies need in-depth investigation. Moreover, the reported negative effects are largely based on epidemiological studies conducted mostly in low-income settings with high childhood mortality. This highlights the need for more mechanistic studies, including in high-income settings with low child mortality, to refute or confirm these observations. A systematic review sanctioned by WHO on off-target effects concluded that the existing epidemiological evidence was not robust enough to support or reject deleterious heterologous effects conferred following DTP vaccination [64]. A Gambian study showed that vaccinating infants with DTP decreases both innate-pro-inflammatory responses to TLR4 stimulation and overall T-cell immunity in females, providing a mechanistic explanation as to why this vaccine might increase susceptibility to other pathogens [65]. The study further demonstrated downregulation of expression of type 1 interferon gene expression following DTP immunisation [65]. It is not clear whether similar findings would be applicable to low-mortality-high income settings considering that most, if not all observations on non-live vaccines, emanate from high mortality-low-income regions. Further immunological evidence that directly links with observations from epidemiological studies in both high and low-income settings would be required to comprehensively address this uncertainty.

Vaccine timing is another important factor that requires consideration in order to optimise off-target vaccine effects. Indeed, it has been proposed that neonatal targeted vaccination within the first days of life as opposed to infant vaccination has the potential to modulate the infant immune system and boost immunity from birth. BCG, OPV and Hepatitis B vaccine administration during the first days of life in some countries has previously shown to be a successful and a practical means of boosting immune responses in newborns [66]. Apart from modulating the infant immune system, potential additional benefits include early protection against infections by shortening the vulnerability period where newborns are more susceptible to a range of pathogens [66]. Additionally, administration of vaccines within the first days of life appears advantageous in achieving good vaccine coverage given that healthcare services are conveniently more accessible at birth and easily implementable [66]. This would fit better in resource-constrained settings where access to healthcare systems is compromised.

## 5. Sex Differential Off-Target Effects

The off-target effects of vaccines have been shown to be sex-differential in multiple studies. Generally, negative non-specific effects induced by non-live vaccines such as DTP appear to be more pronounced in females than males [19,61,67,68]. A study by Nohoh et al. (2016) demonstrated that DTP-vaccinated females had overall suppressed T-cell reactivity and innate pro-inflammatory responses and showed decreased αCD3/αCD28-stimulated type 1 cytokines responses. Contrarily, live-attenuated vaccines such as smallpox, BCG and measles vaccines have shown to provide more beneficial effects in females than in males under the age of five [15,16,19,67,68,69]. Indeed, the same is true for the targeted effects of vaccines with females which were found to elicit more robust antibody-specific responses following vaccination as compared to males for multiple vaccines [66,69]. Immunological sex differences have so far been tied to inherent variations such as hormonal and X-linked alterations that have been shown to be evident during puberty and infancy [67,69].

The mechanisms driving the observed sex differences in off-target effects of vaccination are yet to be fully elucidated. However, X-linked immune response genes and sex hormones have been suggested to play a role in these vaccine-induced sex differential effects [70]. Low level expression of the X chromosome-encoded TLR7 gene on B cells in influenza-vaccinated males compared to females appears to contribute to the sex differential vaccine-induced responses in mice [19,71]. The steroid hormone milieu has also been shown to affect vaccine-induced antibody responses in males and females of both humans and mice [71]. Harnessing this information could inform development of vaccines that improve vaccine-specific responses and off-target beneficial effects in a sex-specific manner.

## 6. Current Rotavirus Vaccines

Early observations that children with secondary exposure to rotavirus were protected from moderate-to-severe rotavirus-associated gastroenteritis paved the way for rotavirus vaccine development [72]. The first ever rotavirus vaccine to be approved was the human-rhesus reassortant live-attenuated tetravalent rotavirus vaccine (RotaShield^®^, Wyeth-Lederle, Marietta, PA, USA). RotaShield^®^ usage was discontinued nine months after rollout in the USA in 1999 due to a reported increased risk of intussusception [5,73,74]. Currently, four live-attenuated rotavirus vaccines are WHO prequalified for global use (Table 1) and 114 countries have included rotavirus vaccines in their national or sub-national routine immunisation programme. Rotarix^®^ (GlaxoSmithKline, Rixensart, Belgium), a monovalent G1[P8] live-attenuated vaccine and RotaTeq^®^ (Merck, Whitehouse Station, NJ, USA), a live-attenuated pentavalent (G1, G2, G3, G4, [P8]) bovine-human reassortant vaccine, are currently the most widely used rotavirus vaccines [5,6]. Rotavac^®^, G9[P11]-(Bharat Biotec, New Delhi, India) and Rotasiil^®^, G1, G2, G3, G4, G9 (Serum Institute, Pune, India) have been introduced into the National Immunisation program in India with introductions planned in Benin (Rotavac^®^) and Democratic Republic of Congo and Burkino Faso (Rotasiil^®^). The Lanzhou Lamb (G10P [12]) and Rotavin-M1^®^ (G1P[8]) are available in the market for use in China and Vietnam, respectively. Over 80 countries have incorporated Rotarix^®^ into their immunisation programmes, 16 countries have introduced RotaTeq^®^ and 10 countries have introduced both Rotarix^®^ and RotaTeq^®^ into their routine immunisation schedules. Whilst both vaccines are administered to infants from 6 weeks of age, Rotarix^®^ is given in two doses and RotaTeq^®^ is administered in a three-dose schedule [5], (Table 1).

Rotarix^®^ is currently the most widely used rotavirus vaccine globally [6]. It was initially introduced in Mexico in 2006 and by the end of 2010, mortality due to diarrhoea had decreased by over 40% in Mexican children under the age of five years [6,75]. The decline in diarrhoea mortality could not be explained by specific vaccine-induced immunity alone. Indirect protection due to a reduction in transmission of rotavirus from vaccinated children was cited as an attributable factor for the unexpectedly large decrease in diarrhoea mortality. In contrast, a study in Finland reported a lack of evidence to support reduced non-rotavirus acute gastroenteritis following vaccination with RotaTeq^®^ [76]. The study suggested differences in geographical settings as a possible contributing factor to the conflicting results [76]. These findings support further investigations into the mechanistic explanation of off-target effects in the reported sites and the ideal population that would benefit from these effects. Prospective birth cohort studies from Malawi on mortality impact and vaccine effectiveness of sequential 13-valent pneumococcal conjugate (PCV13) and monovalent rotavirus vaccine RV1 provided population-level evidence of infant mortality reduction further highlighting the importance of increased vaccine coverage in high-mortality settings [77].

## 7. The Need to Improve the Performance of Rotavirus Vaccines

Despite significant strides established so far following rotavirus vaccine introduction in many countries, rotavirus vaccines have generally underperformed in high mortality settings. Bergman et al. (2021) reported that Rotarix prevented 93% of severe rotavirus diarrhoea cases and 52% of severe all-cause diarrhoea cases in low-mortality settings. In high mortality settings, Rotarix prevented 58% of severe rotavirus diarrhoea cases and 27% of severe all-cause diarrhoea cases [78]. Underperformance of rotavirus vaccines particularly in low-income countries is of great concern which warrants development of alternative vaccines and new approaches tailored to maximise vaccine protective effects. High levels of breast milk antibodies, transplacentally transferred maternal antibodies, and a high prevalence of malnutrition and enteropathy are some of the factors that are thought to be associated with reduced rotavirus vaccine effectiveness in low-income settings [79,80]. The neonatal live-attenuated rotavirus vaccine RV3-BB (Biofarma, Bandung, Indonesia), nonreplicating P2-VP8-P[8] and P2-VP8-P[4]P[6]P[8] subunit rotavirus vaccine (NRRV, PATH), virus-like particle vaccines (Baylor College of Medicine), inactivated whole virus vaccine (Curevac) and RNA-based vaccines (CDC) [5,6,81] are alternative rotavirus vaccines currently in development (Figure 1).

The persistence of a high rotavirus disease burden in early life requires development of better vaccines that can be administered as early as the first week of life. The current licenced rotavirus vaccines are administered from 6 weeks of age, therefore missing the opportune window when the neonatal immune system is more susceptible to infections [6]. RV3-BB is an oral live naturally attenuated rotavirus vaccine developed at the Murdoch Children’s Research Institute, Australia [6]. RV3-BB is based on the human neonatal rotavirus strain RV3 (G3P[6]) associated with asymptomatic infection of the neonatal gut. Its ability to offer early protection in the presence of possible inhibitory effects from maternal antibodies and breast milk is intriguing [82,83,84]. Furthermore, the immaturity of the neonatal gut microbiome and the limited intake of breast milk may present an advantage for a birth dose of rotavirus vaccine. Rotavirus VP4 [P] binding is genotype-dependent and differential expression of histo-blood group antigens (HGBA) could explain the disparity in rotavirus vaccine efficacy between populations. For P[8] infections, including the P[8]-based Rotarix vaccine, secretor-positive individuals are more susceptible to infection and have higher vaccine shedding, than observed in secretor-negative individuals. In contrast, there was no difference in vaccine take after administration of RV3-BB vaccine, based on a P[6] strain, observed by secretor status, by Lewis status or by combined secretor or Lewis status. This intrinsic feature may mean that RV3-BB has the potential to provide improved protection in Africa, where the Lewis-negative phenotype is more prevalent (20–35% in some countries) and where P[6] rotavirus strains causing disease are endemic [85,86,87].

Rotavirus vaccines are routinely administered with the first dose at 6 weeks of age. Developed from a human neonatal rotavirus strain, RV3-BB vaccine is an ideal candidate for a neonatal administration strategy, with the first dose administered at or soon after birth. RV3-BB phase II clinical trials conducted in both a high-income, low-child mortality country (New Zealand) and a low-middle income, high-child mortality country (Indonesia and Malawi) have shown that RV3-BB is immunogenic and well tolerated when administered as a three-dose neonatal or a routine infant schedule [83,84,88,89]. Vaccine efficacy against severe rotavirus gastroenteritis among Indonesian infants after three doses of RV-BB in the neonatal schedule was 94% at 12 months of age and was 75% at 18 months of age [84]. In Malawian infants, anti-rotavirus IgA seroconversion was observed in 41.3% to 58.8% of participants 4 weeks post RV3-BB vaccination with three doses in the neonatal schedule. A further 62.3% to 71.9% seroconversion was noted in participants 18 weeks post RV3-BB vaccination with three doses in the neonatal arm [89].

## 8. Future Considerations

Clearly, the concept of off-target effects of vaccines continues to evolve and important lessons are yet to be unmasked. However, there are important questions that remain to be addressed in order to utilise this intrinsic feature of vaccines to inform vaccine design. To date, limited studies have analysed the putative off-target beneficial effects of live-attenuated rotavirus vaccines either from an epidemiological or mechanistic point of view [90,91,92]. Moreover, it is not clear whether live-attenuated rotavirus vaccines incorporated into immunisation programmes would influence the performance of other vaccines known to induce off-target effects, such as BCG. Robust controlled studies that investigate how a live-attenuated rotavirus vaccine administered at birth could impact off-target effects and BCG are required.

While the current understanding of mechanisms of vaccine-induced off-target effects is largely drawn from BCG studies, it would be intriguing to investigate the possible genes and signalling pathways that may be involved in influencing innate and non-targeted adaptive responses following rotavirus vaccination. Animal model studies by Lee et al. (2018) demonstrated that a cold-adapted live-attenuated influenza vaccine (CAIV) provided immediate protection against respiratory syncytial virus (RSV) infection. They further reported that TL3/TL7 signalling pathways were essential for these observed rapid beneficial off-target effects [93,94,95,96]. Rotavirus, like other mucosal viruses such as RSV and influenza viruses share common features despite them belonging to different virus families. For instance, infection with these viruses triggers an early innate immune response that attempts to clear infection within the mucosal sites. These viruses also commonly induce innate immunity via PRRs such as TL3, TL7 and retinoic acid-inducible gene I (RIG-I). Rotavirus similarly triggers innate sensors including RIG-I, TLR2, TL3, TLR5, TL7 and innate immune cells such as dendritic cells and natural killer cells dominantly in humans and to a lesser extent mouse models [97,98,99,100]. It is possible that some or all of the established sensors and innate cells would be plausible mediators of immunomodulatory effects by live-attenuated rotavirus vaccines. Given the availability of advanced systems biology techniques at our disposal, off-target effects of live-attenuated vaccines can be explicitly studied using approaches such as epigenetics, metabolomics, proteomics and transcriptomics [11]. In vitro stimulation assays and immunophenotyping are also useful approaches to studying vaccine off-target effects [18,101].

Efforts to improve rotavirus vaccine effectiveness in low-income countries include development of next generation oral rotavirus vaccines, exploration of vaccine adjuvanticity, alternative vaccine administration schedules and alternative routes of vaccine administration such as injectable subunit or inactivated rotavirus vaccines. It is hoped that these alternative options will address some of the challenges associated with oral vaccination in children in low-income countries including oral vaccine interference, negative impact of enteropathy and burden of enteric infections. However, when considering these alternative approaches the potential for impact on non-specific effects currently provided by an oral live-attenuated rotavirus vaccine should be assessed [6,80]

Neonates have unique immune systems that are susceptible to a range of infections largely due to limited experience and exposure to microbes. The neonatal immunological milieu favours inhibition of T helper 1 (Th1) and pro-inflammatory responses that further impair innate and adaptive responses to neonatal diseases [66,102]. Furthermore, it has been shown that a high frequency of suppressive regulatory T cells that induce increased levels of the inhibitory cytokines IL-10 and TGF-β characterise the developing neonatal/infant immune system [103,104,105]. This inhibitory milieu reportedly limits the expansion of follicular Th1 cells and sequentially diminishes germinal centre B-cell responses that are critical for generation of immune memory [66]. Studies of the immunomodulatory effects of live-attenuated rotavirus vaccines, such as RV3-BB, on the neonatal immune system would allow us to gain understanding of the potential off-target effects of this novel and promising live-attenuated vaccine (Figure 2). These effects might confer additional benefits over its protective effects against rotavirus infection with positive implications in vulnerable neonates and infants [106].

A lack of an appropriate research model to study off-target effects in neonates and infants has presented a challenge to its further understanding. Human neonatal trials are often impractical and ethically challenging thereby limiting interrogation of immune responses in early life. Neonatal mouse models provide a practical means of investigating off-target effects and vaccine interventions, however, inherent variations between humans and mice restrain the direct extrapolation of findings to humans [66]. Umbilical cord blood is an alternative means of assessing ex vivo immune responses in human neonates prior to vaccination. Use of neonatal gnotobiotic pig (Gn) in biomedical research is yet another valuable model and is so far a proven model of human rotavirus disease [82,107]. Pigs and humans share high genomic and protein sequence, thus facilitating extrapolation of findings to humans [108].

Umbilical cord blood samples can be leveraged to provide useful insights into the immune ontogeny and the suspected vaccine influence in inducing off-target effects of vaccines [109]. A comparison between cord blood sample before and after vaccine intervention could uncover useful insights in the context of potential off-target effects induced by vaccines administered within the first days of life. It is therefore imperative that more mechanistic studies of off-target effects are conducted for live-attenuated rotavirus vaccines in order to generate evidence that could inform understanding of any beneficial off-target effects and lead to improved vaccines that are tailored to the unique neonatal and infant immune system.

Important mechanistic questions that require further attention include:How long do vaccine-induced off-target effects last?To what extent do external factors such as maternal vaccine exposure, birth method and vaccine type influence occurrence of off-target effects?What mechanistic approaches can separate vaccine-induced off-target effects versus environmentally induced off-target effects (such as early pathogen exposure, infant gut microbiome, maternal vaccine/pathogen exposure)?At what age does sex influence the occurrence of off-target effects?Where adequate evidence on sex interaction with vaccine-induced effects is gathered, what is the practicability of developing vaccines tailored to specific sex or about interindividual variations that occur naturally?In the context of a neonatal live-attenuated RV3-BB vaccine, can co-administration of RV3-BB with known off-target inducers such as BCG affect induction of anticipated off-target effects?

As the hunt for improved knowledge and understanding of vaccine-induced off-target effects continues, interpretation of findings must take into consideration the potential cofounders such as the geographical or epidemiological setting, maternal vaccine or pathogen exposure, biological factors such as sex, disease burden to infections such as norovirus, cytomegalovirus and vaccine timing (Figure 3). Attributing changes in morbidity or mortality to vaccine effects is often challenging, especially in high mortality low income setting with concurrent and changing public health and social-welfare interventions. Therefore, robust study designs which use relevant quality data to accurately quantify indirect vaccine effects and overall vaccine impact are essential before estimating any off-target effects in the real world.

In summary, live-attenuated rotavirus vaccines have the potential to induce beneficial off-target effects which could be clinically significant when administered to the immunologically naïve neonate. The live-attenuated neonatal rotavirus vaccine, RV3-BB, administered from birth offers potential to provide early protection against severe rotavirus disease but also infection from other pathogens via its off-target effects on the immune system of the newborn.

## Figures and Tables

**Figure 1 vaccines-10-00418-f001:**
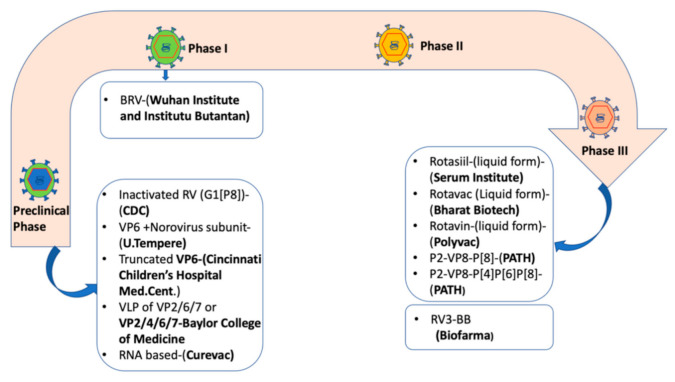
Rotavirus vaccine candidates in development as of March 2021 [5,6]. Abbreviations: BRV = bovine rotavirus pentavalent vaccine, VLP = virus-like particle, RV = Rotavirus vaccine, CDC = Centre for Disease Control.

**Figure 2 vaccines-10-00418-f002:**
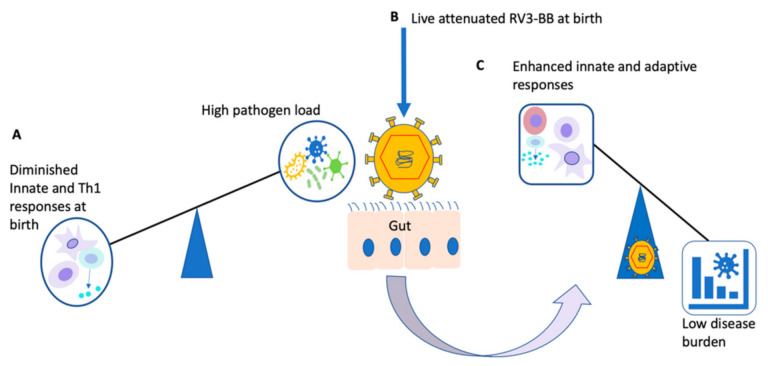
Illustration of potential beneficial off-target effects of live-attenuated RV3-BB vaccine: At birth and in younger infants, the immune system generally favours inhibition of Th1 and innate responses (**A**). Live-attenuated vaccines such as RV3-BB administered at birth (**B**) would circumvent the imbalance towards enhanced immune responses (**C**) [30,100,102].

**Figure 3 vaccines-10-00418-f003:**
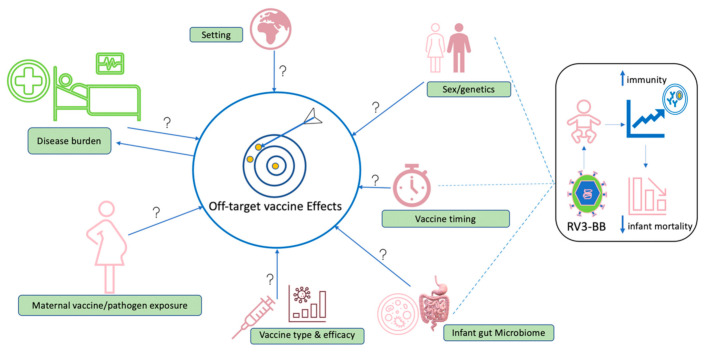
Potential areas that require further study in the context of off-target effects of vaccination. Factors with potential to influence induction of off-target effects include geographical setting, sex/genetics, vaccine type, performance and vaccine co-administration, vaccine timing, composition of gut microbiome, prior maternal exposure to vaccine or microbes and disease burden in endemic settings. We hypothesise that a live-attenuated rotavirus vaccine (RV3-BB) may cause the development of beneficial off-target vaccine effects when administered at birth, with the potential to interact with sex and infant gut microbiome contributing to overall reduced infant mortality.

**Table 1 vaccines-10-00418-t001:** Current rotavirus vaccines.

Product Name	Manufacturer	Composition	Doses	Route of Administration	Regulatory Status
Rotarix^®^	GSK	G1P[8]-Live attenuated	2	Oral	WHO prequalified
Rotavac^®^	Bharat Biologicals	G9[P11]-Live attenuated	3	Oral	WHO prequalified
RotaTeq^®^	Merck	G1, G2, G3, G4, P[8]-Live attenuated	3	Oral	WHO prequalified
Rotasiil^®^	Serum Institute of India	G1, G2, G3, G4, G9-Live attenuated	3	Oral	WHO prequalified
Rotavin-M1^®^	POLYVAC	G1P[8]-Live attenuated	3	Oral	Nationally licensed
Lanzhou Lamb^®^	Lanzhou Institute of Biological Products	G10P[12]-Live attenuated	1	Oral	Nationally licensed

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
