# Peer review of "Leveraging Beneficial Off-Target Effects of Live-Attenuated Rotavirus Vaccines"

_vaccines, 2022, doi:10.3390/vaccines10030418_

Round 1

Reviewer 1 Report

This is a nice and well-written review article. 

However, I am concerned about chapter 4 that give the impression that inactivated vaccine have negative off-target effects while live vaccines positive. I am aware that there are examples for this from epidemiological studies but the data is not sufficient to give this general statement. Another point of concern is that most or all reported negative off-target effects were from the group of Peter Aaby. It does not mean that they are wrong but given the importance of vaccination, it is important to carefully address these issues with future immunological and epidemiological investigations. Furthermore, the negative off-target effects were, to my knowledge, only observed in high children mortality regions and again general conclusions are too risky at this point. I suggest therefore to keep the focus on the rotavirus vaccines, and simply mention that off-target effects could also possible citing e.g. an epidemiological study and mentioning DTP as an example. 

Also, the sex differences that are addressed are citing 4 review articles by one of the authors (66-68). Either this review simply refers that the topic has been reviewed elsewhere or cites the original works.

Author Response

Dear Reviewer 1,

We are grateful for your  constructive and valuable comments that have enhanced the value of our  manuscript. We have made the requested corrections in the attached detailed list of our Point-by-point responses.

Many thanks,

Khuzwayo 

Reviewer 2 Report

Please see attached the comments.

Author Response

Dear Reviewer 2,

We are grateful for your  constructive and valuable comments that have enhanced the value of our  manuscript. We have made the requested corrections in the attached detailed list of our Point-by-point responses.

Many thanks,

Khuzwayo 

Round 2

Reviewer 2 Report

No additional comment.

Author Response

No further suggestions made by the reviewer so no point-by-point response provided.